# Multiple Intermediary Model Test of Adolescent Physical Exercise and Internet Addiction

**DOI:** 10.3390/ijerph20054030

**Published:** 2023-02-24

**Authors:** Cheng Qiu, Yufei Qi, Yao Yin

**Affiliations:** 1Police Sports and Warfare Training Academy, People’s Public Security University of China, Huangyi Rd., Daxing District, Beijing 100038, China; 2Department of Physical Education and Research, Central South University, 932 Lushan South Rd., Changsha 410083, China; 3College of Cross-Cutting Education, Beijing College of Finance and Commerce, No. 15, Shuuiduizi East Rd., Chaoyang District, Beijing 100026, China

**Keywords:** adolescents, physical exercise, self-efficacy, self-control, psychological resilience, Internet addiction

## Abstract

On the basis of self-efficacy theory, self-control theory and psychological resilience theory, this paper discusses the relationship between physical exercise, self-efficacy, self-control, psychological resilience and Internet addiction among adolescents in Beijing. A convenience sampling method was used to conduct a questionnaire survey on physical activity and Internet addiction among 466 adolescents from first to third year in 10 high schools in Beijing, of which 41% were girls and 59% were boys; 1.9% of students were 14 years old, 42.5% were 15 years old, 23.4% were 16 years old, 31.3% were 17 years old and 0.9% were 18 years old. Using the research methods of the literature, correlation analysis and multiple intermediary structure model, this paper constructed and tested the multiple intermediary model between physical exercise and Internet addiction. The results show that physical exercise can significantly predict self-efficacy, psychological resilience and self-control; self-efficacy, psychological resilience and self-control significantly interfered with Internet addiction behaviour; there was a significant difference in the total effect of multiple intermediaries; the effect value was −0.173; the specific indirect effects of self-efficacy, psychological resilience and self-control had intermediary effects in the relationship between physical exercise and Internet addiction; and there was no difference in specific indirect effects. This paper puts forward some countermeasures and suggestions to prevent teenagers’ Internet addiction from cultivating, such as through good sports activities, thereby improving their Internet addiction. We should actively strive to improve teenagers’ deep understanding of the effect of physical exercise and gradually form physical exercise habits, with sports addiction replacing Internet addiction.

## 1. Introduction

The Law of the People’s Republic of China on the Promotion of Family Education stipulation describes that parents or other guardians of minors shall reasonably arrange a time for physical exercise for minors and prevent them from becoming addicted to the Internet [1]. With the continuous development of Internet users towards younger ages, Internet addiction among secondary school students has become an increasingly serious global problem. Adolescents’ addiction to the Internet can lead to a series of problems: losing interest in learning, affecting academic progress and destroying family harmony. It is, therefore, a daunting and urgent task to prevent and control youths’ addiction to the Internet. The Urgent Notice on Preventing Primary and Secondary School Students from Becoming Addicted to the Internet emphasises the importance of parental education and family prevention, which also highlights effective communication and exchange between parents and students. Pei Tao, working for the Mental Health Education and Consultation Centre at Nanjing Normal University, suggests that parents should cultivate healthy and broad interests for their children and “broaden the interest of health and cultivate the discipline of liberalism” [2]. Adolescents’ cognitive development is still on the way, and their self-control, mental toughness and self-efficacy are weak, but related studies have shown that physical exercise positively affects self-control [3], mental toughness and self-efficacy [4]; self-control [5], mental resilience [6] and self-efficacy [7] negatively predict mobile game addiction. Therefore, on the basis of promoting the healthy and rational use of the Internet among adolescents and safeguarding their healthy growth, this study explores the multiple mediating effects of self-control, self-efficacy and mental resilience on the relationship between physical exercise and Internet addiction by constructing a multiple mediating conceptual model of the relationship between physical exercise and Internet addiction to provide a theoretical framework and practical guidance for optimising the healthy development of the family educational environment.

### 1.1. Theoretical Assumptions and Conceptual Models

#### 1.1.1. The Intermediary Effect of Self-Efficacy on Physical Exercise and Internet Addiction

Self-efficacy is one’s own obsession with his or her strength in successfully completing a task and is one of the strongest predictors of motivational factors, which can also conjecture an individual’s deliberate behaviour in any situation [8]. Increased self-efficacy is influenced by physical exercise [9], and participating in physical exercise increases adolescents’ levels of perceived self-competence, which enhances confidence and expectations to complete sports activities and increase their self-efficacy in sport. Increased self-efficacy improves the ability and beliefs needed to achieve one’s behavioural goals, i.e., the belief that “I can do it” is intensified. Increased self-efficacy has a negative effect on Internet addiction [10]. An empirical study by Sheng et al. measured 1084 secondary school students and found a significant positive association between physical activity and self-efficacy [11]. Berte et al. examined the relationship between Internet use and self-efficacy in a study of 505 Palestinian university students and found a high negative correlation between Internet addiction use patterns and self-efficacy [12]. As individuals actively strive to achieve the behavioural goals they set for themselves, they reduce their dependence on the Internet. Self-efficacy becomes a key factor in bridging physical exercise and Internet addiction. Therefore, the following hypotheses are proposed: H1—physical exercise positively affects self-efficacy; H2—self-efficacy significantly and negatively affects Internet addiction; and H3—self-efficacy has a significant intermediary effect on physical exercise and Internet addiction.

#### 1.1.2. Intermediary Effects of Mental Resilience on Physical Exercise and Internet Addiction

Mental resilience refers to the personal experience of thriving and excelling through hardship, rather than simply returning to normal functioning [13]. Physical exercise can improve the mental resilience of physically disadvantaged students in university by increasing their level of physical fitness [14], and sustained and moderately frequent physical exercise can increase an individual’s level of mental resilience and motivate them to actively adapt to physical exercise habits. The unique characteristics of physical activity, such as recreation, competition and openness, provide strong support for the development of the individual’s psychological resilience level [15]. The individual will be challenged to learn new knowledge and skills during physical activity, and in the process of overcoming the challenges, the individual’s psychological changes will continue to break through the challenges and contribute to an improvement in the individual’s psychological resilience [16]. It has been shown that physical activity has a positive effect on the level of psychological resilience. People with high psychological resilience are able to intervene in their own behavioural problems through the regulation of their own abilities, and psychological resilience plays an important role in intervening in behavioural problems [17]. Some studies have shown that the main reason for mobile phone addiction among university students is the fear of missing out on what is “trending” on the Internet and the need to check social media status and videos at all times, which in turn increases their dependence on mobile phones [18,19]. This study suggests that individuals with good psychological resilience will self-regulate in the face of the Internet to reduce the impact of the Internet and avoid addictive behaviours. The environment that individuals with high mental resilience behaviours offer helps to counteract the online environment and predicts more positive self-control, which can effectively intervene in the development of online addictive behaviours and contribute to the dynamic adaptation process of the adolescent individual to the surrounding environment. Mental resilience becomes a key variable in linking physical exercise and Internet addiction. Therefore, the following hypotheses are proposed: H4—physical exercise positively affects mental resilience; H5—mental resilience significantly negatively affects Internet addiction; and H6—mental resilience has a significant intermediary effect on physical exercise and Internet addiction.

#### 1.1.3. Intermediary Effects of Self-Control on Physical Exercise and Internet Addiction

Self-control is the ability of individuals to make adaptive adjustments to their cognition, emotions and behaviours, according to the requirements of goals and tasks [20], which can prevent impulsive behaviours and help to resist temptations and other behavioural patterns. Physical activity significantly and positively predicts self-control in adolescents, and there is a strong relationship between physical activity and self-control. This confirms the close relationship between physical activity and self-control. Exercise psychology suggests that physical exercise is an effective way to enhance greater self-control [21,22], and different modalities of physical exercise, such as aerobic exercise and endurance training, can promote self-control [23]. Physical exercise can be effective in improving self-control [24]. Self-control can also significantly predict Internet addiction, with lower self-control associated with higher levels of addictive Internet behaviour [25]. Self-control has a significant negative predictive effect on adolescent Internet addiction, with the stronger the individual’s self-control, the higher the ability to solve problems and adjust emotions, and the lower the level of Internet addiction. The greater the ability to adapt, the lower the level of Internet addiction [26]. Considering the possible correlation between self-control and Internet addiction, improved self-control may imply a decrease in Internet addiction. Yang et al. (2019) concluded that self-control plays an important mediating role between physical activity and mobile phone dependence [27]. This suggests that self-control acts as a “bridge” between physical exercise and Internet addiction. Accordingly, the following hypotheses are proposed: H7—physical exercise positively affects self-control; H8—self-control significantly negatively affects Internet addiction; and H9—self-control has a significant intermediary effect on physical exercise and Internet addiction.

According to the theoretical hypotheses above, a theoretical model of the relationship between physical exercise and Internet addiction behaviour was constructed (see Figure 1).

## 2. Methodology

### 2.1. Participants

A total of 500 students in 10 grades were randomly selected from 3 grades in 3 high schools in Beijing using a convenience sampling method and tested using a physical exercise and Internet addiction questionnaire. A total of 466 valid questionnaires were collected, with an effective rate of 93%. Of these, 41% were girls and 59% were boys; 35.8% were from rural areas and 64.2% were from towns; 1.9% of students were 14 years old, 42.5% were 15 years old, 23.4% were 16 years old, 31.3% were 17 years old and 0.9% were 18 years old.

### 2.2. Research Instruments

#### 2.2.1. Physical Exercise Rating Scale

The Physical Exercise Rating Scale (PARS-3), revised by Liang Deqing et al. [28], was used, which examines the amount of physical exercise in terms of intensity, time and frequency of participation in physical exercise on a 5-point Likert scale. The corresponding scores are 1–5, and the amount of physical activity score = intensity score x (time score—1) x frequency score, and the higher the score, the greater the amount of physical activity; Cronbach’s Alpha coefficient for the scale in this study was 0.602.

#### 2.2.2. Psychological Capital Scale

The two subscales of six items of self-efficacy and five items of mental resilience from the psychological capital entry developed by Luthans et al. [29] were used on a 5-point Likert scale. The corresponding scores are 1–5, with higher scores indicating higher levels of self-efficacy and psychological resilience, with a Cronbach’s Alpha coefficient of 0.863 for the self-efficacy scale and 0.693 for the psychological resilience scale.

#### 2.2.3. Internet Addiction Scale

The Internet Addiction Inventory (CIAS-R) developed by Chen Shuhui [30] was used, containing five dimensions of compulsive internet surfing, withdrawal response, tolerance, interpersonal relationships and health and time management problems, using a 5-point Likert scale. The corresponding scores are 1–5, with higher scores indicating higher levels of self-control, and Cronbach’s Alpha coefficient for this scale in this study was 0.868.

#### 2.2.4. Self-Control Scales

The self-control scale developed by Tan Shuhua et al. [31] was used, which contains five dimensions: impulse control, healthy habits, resisting temptation, concentration. The scale was designed using the Likert 5-point scale. The Likert 5-point scale was used to design the questionnaire by replacing the reverse scoring items with positive scoring. The corresponding scores are 1–5, with higher scores indicating higher levels of self-control, and Cronbach’s Alpha coefficient for this scale in this study was 0.868.

### 2.3. Mathematical and Statistical Methods

The data collected were entered and collated using IBM SPSS 26.0 (SPSS Inc., Chicago, IL, USA); relevant variables were analysed; a model of physical activity and Internet addiction was developed using AMOS 24.0 software (SPSS Inc., Chicago, IL, USA); model fit tests, direct effect path tests and, finally, multiple mediated effect tests using bootstrap and whether there were differences in specific indirect effects were tested.

## 3. Results

### 3.1. Common Method Control and Inspection

To prevent possible common method bias in the collected data, the administration process was controlled accordingly. The Harman one-way method was used to test for common bias. The results showed that there were 11 factors with eigenvalues greater than 1, and the variance explained by the first factor was 10.828%, which was less than the critical threshold of 40%, and the amount of common variance extracted was higher than 70%, indicating that there was no common method bias [32].

### 3.2. Reliability Tests and Correlation Analysis for Each Variable

#### 3.2.1. Test for Reliability of Each Variable

The data were tested for convergent validity and compositional reliability using SPSS 26.0 (see Table 1), and Cronbach’s alpha was >0.7 for all dimensions except physical exercise and mental resilience, which were >0.6 and acceptable, meeting the criteria recommended by Hair [33]. The CR of the component reliability ranged from 0.792 to 0.923 (>0.5), indicating that the scale has good convergent validity and component reliability. The AVE method was proposed by Fornell and Larcher. The average variance extracted for each dimension needs to be greater than the squared value of the correlation coefficient between each dimension and the dimension, and because AVE is a squared value, it must be converted to the same squared units before it can be compared with Pearson’s correlation. The bold diagonal lines in this study (Table 2) are AVE open root values, all of which are greater than the off-diagonal correlation coefficients, so the variables have differential validity.

#### 3.2.2. Correlation Analysis for Each Variable

Compulsive Internet surfing was significantly associated with frustration tolerance, withdrawal response, relationships and health, time management, health habits, temptation resisting, recreational abstinence, physical exercise, self-efficacy and mental toughness (*p* < 0.01 or *p* < 0.05); withdrawal response was significantly associated with interpersonal relationships and health, time management problems, healthy habits, resisting temptation, entertainment control, physical exercise, self-efficacy and mental resilience (*p* < 0.01 or *p* < 0.05); relationships and health were significantly associated with time management, health habits, impulse control, entertainment control, physical exercise and self-efficacy (*p* < 0.01 or *p* < 0.05); time management was significantly associated with entertainment control, physical exercise and self-efficacy (*p* < 0.01 or *p* < 0.05); impulse control was significantly associated with health habits, temptation resisting, learning concentration, entertainment control, physical exercise and mental resilience (*p* < 0.01 or *p* < 0.05); health habits were significantly associated with temptation resisting, learning concentration, entertainment control, physical exercise and mental resilience (*p* < 0.01 or *p* < 0.05); learning concentration was significantly associated with entertainment control, self-efficacy and mental resilience (*p* < 0.01); physical exercise was significantly associated with self-efficacy and mental resilience (*p* < 0.01); and self-efficacy was significantly associated with mental resilience (*p* < 0.01). The diagonal AVE open root sign scores were all greater than the dimension and dimensional correlations.

### 3.3. Model Goodness-of-Fit Tests

The fit of the physical exercise and Internet addiction equation model constructed in this study suggests a good result. CMIN/DF = 1.82 < 3; GFI = 0.827 > 0.8 acceptable; AGFI = 0.813 > 0.8 acceptable; IFI = 0.913; CFI = 0.912; TLI = 0.908; and RMSEA = 0.042 < 0.08. The above empirical path analysis fit met the general rule-of-thumb criteria suggested by scholars [34]. The results can be used to analyse the statistical data at a later stage, indicating that the structural equation model of physical exercise and Internet addiction is valid. The structural equation model of physical exercise and Internet addiction is shown in Figure 2 below.

### 3.4. Testing of Model Hypotheses

#### 3.4.1. Direct Effect Path Test

The results of the direct pathways for the relationship between physical exercise and Internet addiction (see Table 3) showed that the five latent variables of physical exercise, mental resilience, self-control, self-efficacy and Internet addiction produced six direct pathways of influence (*p* < 0.05), and the unstandardised coefficient results indicated that each pathway of influence was significant and valid. The standardised path coefficients showed that among the direct influence paths of the latent variables on Internet addiction, the absolute value path coefficient of self-control was the largest, with a standardised absolute value path coefficient of 0.15, while the standardised absolute value path coefficients of self-efficacy and mental resilience were both 0.136. Among the direct influence path coefficients of physical exercise as the dependent variable, the standardised coefficient of mental resilience was the largest at 0.643. The analysis of the above data indicates that the direct path coefficients for Internet addiction are significant, and the effect sizes of the path coefficients vary.

#### 3.4.2. Testing the Effects of Multiple Intermediaries

The intermediary variable is closer to the outcome than the predictor variable, and the intermediary variable itself is a causal variable. The indirect effect is an independent variable influencing the dependent variable through the intermediary variable. The way to test the effect of the indirect effect consists of the causal path intermediary effect test and the product of the indirect effect coefficients test. The causal path test [35], in which a and b are statistically significant, means that the intermediary effect holds; Sobel argues that the indirect effect holds requires testing whether the product of the coefficients of the two paths is significant, using normal theory to develop the Sobel Z test to test whether the effect is significant [36]. Sobel t has a major flaw, requiring the sample indirect effect to be constant, but a*b is basically asymmetric (not constant), and the skewness and kurtosis are not zero [37].

In fact, researchers can easily obtain path multiplication coefficients, only to obtain the value but not the standard error, which cannot be calculated. The standard error of the equation used to seek causal steps coefficients is more suitable for use in an intermediary. Complex models, especially, contain multiple variables, and complex methods can only be solved with SEM [38] and more than two intermediary effects tests with MacKinnon [39].

A bootstrap was used to test for multiple intermediary effects and whether there were differences in specific indirect effects. The results of the tests were as follows (see Table 4): the overall effect, indirect effect, bias-corrected and Percentile 95% confidence intervals for the multiple intermediary model of physical activity and Internet addiction did not include 0, indicating that multiple intermediary effects existed and that the model was fully mediated. The absolute value of the effect size for self-control was 0.072, accounting for 42%; the absolute value of the effect size for self-efficacy was 0.064, accounting for 37%; and the absolute value of the effect size for mental toughness was 0.037, accounting for 21%, from which it can be seen that self-control was the most important in the intermediary effect role, followed by self-efficacy, and mental resilience acted as the least effective intermediary. Indirect effect differences comparing bias-corrected and Percentile 95% confidence intervals included 0, indicating that there were no differences in specific indirect effect differences.

## 4. Discussion

### 4.1. Analysis of Direct Effects

#### 4.1.1. Direct Effects of Physical Activity and Other Latent Variables

Physical exercise has the effect of regulating the body’s tension, improving the psychological state, enabling individuals to adapt to the developing environment, improving their cognition, emotional responses and volitional behaviour in a positive state and maintaining normal regulation [40]. Physical activity has a positive and significant effect on psychological resilience. The higher the level of physical activity, the greater the level of psychological resilience. To increase resilience and self-control in adolescents, external factors are the inducement, and the individual is the key. Related studies have proposed that physical activity enhances the production and release of β-endorphins in the body and reduces hormones, such as adrenaline and cortisol, and stimulates cognitive thinking and affective cognition, which can enhance psychological changes such as self-control, mental resilience and self-efficacy in adolescents [41]. Consistent with this study, physical activity positively and significantly predicted factors such as self-efficacy, mental resilience [42] and self-control [43]. Physical activity can improve adolescents’ cognitive development and enhance the accumulation of psychological energy, thus, maintaining a stable state of mind [44].

#### 4.1.2. Direct Effects of Mediating Latent Variables on Internet Addiction

The intermediary latent variable factors—self-efficacy, mental resilience and self-control dimensions—each had a significantly negative predictive effect on Internet addiction. Internet addiction was negatively correlated with self-efficacy, and the results may help schools, families and society to design appropriate Internet addiction prevention programs for adolescents [45]; mental resilience is a predictor of Internet addiction and an effective way to reduce Internet addictive behaviours, and the findings provide a useful basis for the early detection and intervention of Internet addiction [46]; and mental toughness has a significant negative impact on mobile phone addiction. The stronger the mental toughness, the less severe the mobile phone addiction and the more important the role that mental toughness plays in the development of mobile phone addiction in individuals [17]. Lack of self-control is an important factor [47], and online-game players’ self-control can negatively predict online game addiction [48].

### 4.2. Analysis of Intermediary Effects

#### 4.2.1. Analysis of Self-Efficacy Intermediary Effects

The results of the 5000 bootstrap intermediary effect tests for self-efficacy performed in this study show that the effect size of self-efficacy was −0.064, bias-corrected, and the Percentile 95% confidence interval did not contain 0. The test results indicate that the self-efficacy intermediary effect was significant. According to self-efficacy theory, it is known that college students’ adherence to physical activity can enhance self-efficacy [49,50,51], individual behavioural habits may lead to the formation of lifelong habits, and research on improving adolescent self-efficacy may be useful in designing interventions to reduce Internet addiction [12], and empirical studies have shown that self-efficacy can effectively intervene in Internet addictive behaviours [52,53,54]. Family education should promote youth participation in physical activity to enhance youth self-efficacy and address students’ Internet addiction [4]. The significant mediating role of self-efficacy between physical activity and Internet addiction among college students is consistent with the theoretical implications of previous studies [55]. Physical activity mediates the effect of internet influence on college students’ addiction through self-efficacy, and these influences play a protective role against students’ Internet addiction [56]. Therefore, self-efficacy plays a role in mediating the effect of physical activity on the relationship between physical activity and Internet addiction.

#### 4.2.2. Mental Resilience Intermediary Effect Analysis

The results of the 5000 bootstrap intermediary effects test performed for mental resilience in this study show that the effect size of mental resilience was −0.037, bias-corrected, and the Percentile 95% confidence interval did not contain 0. The test results indicate that the intermediary effect of mental resilience was significant. The theory of mental resilience refers to the ability of adolescents to activate protective factors against undesirable influences (Internet addiction, delayed movement and psychological issues). The characteristics of recreation, competition and interest in physical exercise can enhance adolescents’ mental resilience [34,57], a space for developing the mental toughness of young people, and regular physical activity can enhance both the physical fitness of the individual and the level of interpersonal interaction. The difficulties and challenges encountered during physical activity are internalised over time and can be transformed into individual protective factors, helping to improve the resilience of young people and their level of mental toughness, along with diverse environmental adaptations. Additionally, there is a significant negative correlation between mental resilience and Internet addiction [58,59,60], and increased levels of mental toughness can effectively regulate individual behavioural activities and self-regulation in the face of mobile phone problems. The psychological satisfaction that physical activity brings to the individual can effectively inhibit the occurrence of mobile-phone-dependent behaviour, which shows that mental resilience plays an intermediary effect in the relationship between physical exercise and Internet addiction.

#### 4.2.3. Analysis of Self-Control-Mediated Effects

The results of the 5000 bootstrap intermediary effects tests for self-control execution in this study suggest that the effect size of self-control was −0.072, bias-corrected, and the Percentile 95% confidence interval did not contain 0. The test results indicate that the intermediary effect of self-control was significant. The increase in the level of self-control motivation was related to individuals’ correct perception and evaluation of themselves, and adherence to regular physical exercise could improve self-control, which was negatively related to mobile phone dependence [61]. The self-control resource model indicates that self-control increases after the organism overcomes a short period of physical exercise exertion accompanied by an increase in comfort [62] to reduce the likelihood of Internet addiction [63], with a significant increase in self-control and a significant decrease in Internet addiction and time spent online [64]. Sports participation has a significant effect on Internet addiction when mediated by self-control. Physical activity is effectively necessary in an Internet addiction treatment program and has broader psychological and physical benefits than intervention strategies or pharmacotherapy [65]. Sport has a significant negative predictive effect on the trend of Internet addiction. Self-control plays a fully mediating role in the relationship between sports and Internet addiction [57]. This indicates that self-control plays a role in mediating the effect of physical activity on the relationship between physical activity and Internet addiction.

In conclusion, family education is the starting point and root of the whole education, playing a more important role than school education and social education. The family has a great influence on young people’s Internet addiction, and there is a significant relationship between Internet addiction and family parenting. In the absence of realistic interactions, children become obsessed with the internet in order to find support. Optimising family parenting styles is an effective way to stop students’ Internet addiction [66]. Exercise interventions for adolescent Internet addiction are effective in the long term. The mechanism inherent in exercise interventions for Internet addiction is the activation of instincts that lead to the psychological satisfaction of the individual, which in turn displaces the behaviour of Internet addiction [67]. The “addiction replacement” model of Internet addiction prevention is a cost-effective tool.

Overall, this study proposes a multiple mediation model analysis of physical exercise and Internet addiction, which reveals the mechanism of the effect of physical exercise on Internet addiction in adolescents: physical exercise influences the level of Internet addiction in adolescents through self-efficacy, mental toughness and self-control. Therefore, there is a mediating effect between physical exercise and Internet addiction. In the process of participating in physical exercise, adolescents should choose their favourite sports, pay attention to the time and intensity of exercise and persist in completing physical exercise so as to enhance their self-efficacy, mental toughness and self-control in physical exercise and effectively intervene in the occurrence of Internet addiction. There are also shortcomings in this study. Due to the ethical requirements of the survey and the limitations of the research resources, the sample of this survey is mainly young people, excluding both university students and primary school students, and the research results do not yet cover the whole scope of school sport, so it is hoped that more scientific research methods and theoretical perspectives can be explored in subsequent studies to make up for the shortcomings.

## 5. Conclusions

Physical exercise had a positive predictive effect on self-efficacy and mental resilience, while self-control, self-efficacy, mental resilience and self-control were important factors in intervening in Internet addiction. The empirical test showed that the total indirect effect existed with a total effect size of −0.173, and the three path-specific indirect effects of self-efficacy, mental resilience and self-control also had significant effects, with self-control having the highest intermediary effect size, accounting for 42% of the total indirect effect; followed by self-efficacy at 37%; and mental resilience had the smallest intermediary effect size, accounting for 21%. The empirical tests indicate that the 95% confidence intervals for the bias-corrected and Percentile values for the three pathway-specific indirect effects contained 0, and that there was no significant difference in the comparison of the differences in the three pathway-specific indirect effects.

## Figures and Tables

**Figure 1 ijerph-20-04030-f001:**
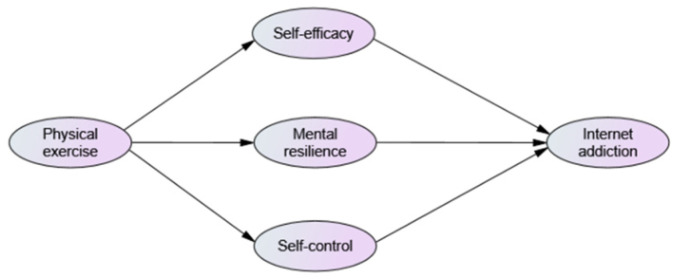
Diagram of the theoretical model of the relationship between physical exercise and Internet addiction.

**Figure 2 ijerph-20-04030-f002:**
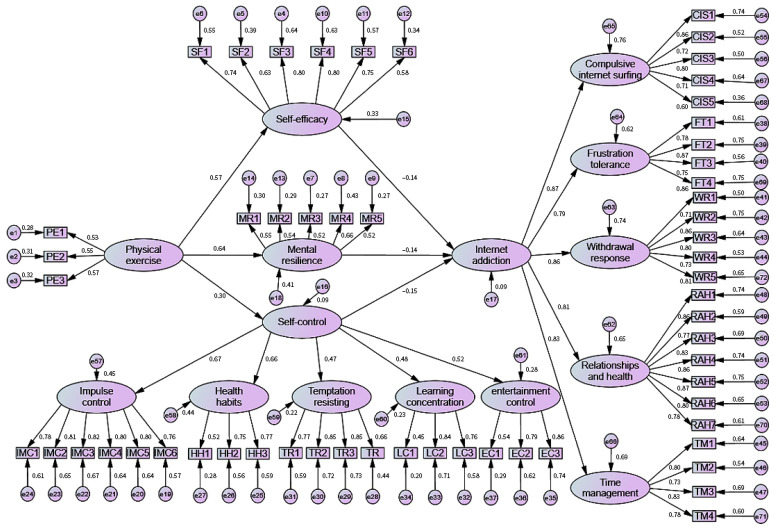
Structural equation model of physical exercise and Internet addiction.

**Table 1 ijerph-20-04030-t001:** Reliability tests for each variable.

Variables	Convergent Validity		Component Reliability
	Cronbach’s Alpha	AVE	CR
Compulsive internet surfing	0.854	0.635	0.897
Frustration tolerance	0.885	0.750	0.923
Withdrawal response	0.880	0.690	0.917
Relationships and health	0.927	0.726	0.949
Time management	0.843	0.713	0.909
Impulse control	0.910	0.690	0.930
Health habits	0.701	0.636	0.839
Temptation resisting	0.857	0.710	0.907
Learning concentration	0.703	0.638	0.839
Entertainment control	0.747	0.679	0.863
Physical exercise	0.602	0.559	0.792
Self-efficacy	0.863	0.598	0.899
Mental resilience	0.693	0.450	0.803

**Table 2 ijerph-20-04030-t002:** Correlation tests for each variable.

	AVE	CIS	FT	WR	RAH	TM	IC	HH	TR	LC	EC	PE	SF	MR
CIS	0.635	**0.797**												
FT	0.750	0.583 **	**0.866**											
WR	0.690	0.693 **	0.596 **	**0.831**										
RAH	0.726	0.649 **	0.582 **	0.628 **	**0.852**									
TM	0.713	0.626 **	0.611 **	0.580 **	0.620 **	**0.844**								
IC	0.690	−0.078	−0.092 *	−0.063	−0.114 *	−0.060	**0.831**							
HH	0.636	−0.133 **	−0.140 **	−0.162 **	−0.101 *	−0.065	0.369 **	**0.797**						
TR	0.710	−0.112 *	−0.118 *	−0.094 *	−0.088	−0.044	0.312 **	0.277 **	**0.843**					
LC	0.638	−0.002	−0.026	−0.057	−0.025	−0.027	0.291 **	0.254 **	0.190 **	**0.799**				
EC	0.679	−0.102 *	−0.063	−0.091 *	−0.101 *	−0.105 *	0.341 **	0.309 **	0.178 **	0.293 **	**0.824**			
PE	0.559	−0.184 **	−0.138 **	−0.159 **	−0.165 **	−0.134 **	0.095 *	0.130 **	0.154 **	0.064	0.029	**0.748**		
SF	0.598	−0.183 **	−0.102 *	−0.167 **	−0.105 *	−0.120 **	0.029	0.092 *	0.033	0.126 **	0.028	0.367 **	**0.773**	
MR	0.450	−0.185 **	−0.092 *	−0.206 **	−0.134 **	−0.087	0.201 **	0.226 **	0.088	0.201 **	0.087	0.336 **	0.34 **	**0.671**

Note: CIS—compulsive internet surfing; FT—frustration tolerance; WR—withdrawal response; RAH—relationships and health; TM—time management; IC—impulse control; HH—health habits; TR—temptation resisting; LC—learning concentration; EC—entertainment control; PE—physical exercise; SF—self-efficacy; MR—mental resilience. ** At the 0.01 level (two-tailed). * Correlations significant at 0.05 level (two-tailed). Diagonal bold text is open root value of AVE. Lower triangle is Pearson’s correlation of dimensions.

**Table 3 ijerph-20-04030-t003:** Results of the path analysis.

Direct Path Impact	Unstd. Path Coefficients	Std. Error	Z	C.R.	*p*	Std. Path Coefficients
Physical exercise -> self-efficacy	0.662	0.090	7.36	7.37	0.000	0.574
Physical exercise -> self-control	0.173	0.047	3.68	3.696	0.000	0.304
Physical exercise -> mental resilience	0.498	0.078	6.38	6.376	0.000	0.643
Self-efficacy -> Internet addiction	−0.096	0.041	−2.34	−2.351	0.019	−0.136
Self-control -> Internet addiction	−0.215	0.092	−2.34	−2.329	0.020	−0.150
Mental resilience -> Internet addiction	−0.144	0.069	−2.09	−2.098	0.036	−0.136

**Table 4 ijerph-20-04030-t004:** Results of the multiple mediation test.

Variable Path	Point Estimation Value	SE	Bootstrapping			
			Bias-Corrected 95%CI		Percentile 95%CI	
			Lower	Upper	Lower	Upper
Indirect effects						
Physical exercise -> self-efficacy -> Internet addiction	−0.064	0.033	−0.147	−0.010	−0.139	−0.006
Physical exercise -> self-control -> Internet addiction	−0.072	0.044	−0.183	−0.004	−0.176	−0.001
Physical exercise -> mental resilience -> Internet addiction	−0.037	0.022	−0.099	−0.007	−0.090	−0.004
Total effect	−0.173	0.052	−0.296	−0.091	−0.292	−0.089
Difference comparison						
Self-efficacy–self-control	0.008	0.062	−0.11	0.141	−0.108	0.144
Self-control–mental resilience	−0.035	0.051	−0.147	0.057	−0.145	0.057
Mental resilience–self-efficacy	0.027	0.039	−0.047	0.11	−0.048	0.107

Note: 5000 self-help samples.

## Data Availability

Not applicable.

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
