# Peer review of "Multiple Intermediary Model Test of Adolescent Physical Exercise and Internet Addiction"

_ijerph, 2023, doi:10.3390/ijerph20054030_

Round 1

Reviewer 1 Report

I was pleased to read the manuscript entitled "Multiple intermediary model test of adolescent physical exercise and Internet Addiction" and to review it.

The study examined the relationship between physical exercise, self-efficacy, self-control, psychological resilience and Internet addiction among adolescents in Beijing. It also provide a theoretical framework and practical guidance for optimizing the healthy development of the family educational environment. From a scientific point of view, the article did not reveal particularly new regularities, but it was interesting to read it as is one of the few studies that was conducted in the selected population and using modern methods of analysis. The article is written in a typical format and is well structured.

Title and abstract – the title is informative and it accurately reflect the manuscript. The abstract is more or less complete and adequately reflects the content of the manuscript. However, I would recommend to note the design of the study and further characterize the studied population (number of subjects, their age), the method of data collection. The list of keywords should be supplemented with the term "adolescents".

Introduction – the Introduction provide sufficient theoretical background for the study. All examined research questions and/or hypotheses were introduced The introduction is structured logically and the text is fluent. The rationale of the study is well described and the study problem is stated clearly. Relevant and unbiased literature was used.

Method – sampling and measures are shortly described and are appropriate to answer the proposed research questions. Here I recommend changing the term "factor" to the term "item". The statistical analysis is described too narrowly, so some results are unclear.

Results – in general, results are clearly organized and presented. However, several inaccuracies were noted:

a) Section 3.1: Please indicate which scale was tested. If multiple scales were tested, report data for each scale separately.

b) Section 3.2.2: It is unclear how the values of unobservable variables were estimated on the basis of items. Please explain this in Methods.

Discussion – the structure of the Discussion is clear. The interpretations is appropriate and is supported by the results. The study findings are discussed with relevant literature. However, the contribution of the study to the field is weakly explained - it is recommended to emphasize the novelty of the study and to indicate its limitations.

General comments:

a) The format of the article (tables, figures) should be more in line with the requirements of the journal (see template of the journal).

b) References must be provided in accordance with the requirements of the journal.

c) Revise text for spelling errors and other inaccuracies.

Thank you for considering my opinion. I encourage authors to keep on working to improve the manuscript.

Author Response

Dear Reviewer, Thank you for your valuable comments. I have completed the revise point by point as your requested, please see the attachment for details.

Reviewer 2 Report

Introduction

You provide some literature but I suggest to add more studies.

You don’t provide much reason to suggest the direction of your hypothesis.  Please specify the theorical basis of your study.

Measures:

Please include validity and reliability data for the measures in the text, including internal consistence.

Results

Table 2 is not clear enough

Limitations

Please, mention limitations 

Discussion

Please better support the discussion in line with your the results and the recent literature

Author Response

(The authors gave the same response as above.)

Round 2

Reviewer 2 Report

The authors have answered to all my previous criticisms.